# An Anomalous Phylogenetic Position for *Deraiotrema platacis* Machida, 1982 (Lepocreadiidae) from *Platax pinnatus* on the Great Barrier Reef

**Rodney A. Bray** [1,*], **Scott C. Cutmore** [2]  **and Thomas H. Cribb** [2]

1   Department of Life Sciences, Natural History Museum, Cromwell Road, London SW7 5BD, UK
2   School of Biological Sciences, The University of Queensland, St Lucia 4072, Australia
*   Correspondence: rab@nhm.ac.uk; Tel.: +44-137-651-2266

**Abstract:** The monotypic genus *Deraiotrema* Machida, 1982 has only been reported once, from the orbicular batfish *Platax orbicularis* (Forsskål) in the waters around Palau in Micronesia (Machida, 1982). It has a body-shape similar to other lepocreadiids from batfishes, such as species of *Bianium* Stunkard, 1930 and *Diploproctodaeum* La Rue, 1926, but differs in having multiple testes in ventral and dorsal layers. Here we report *Deraiotrema platacis* Machida, 1982 for just the second time, infecting the dusky batfish *Platax pinnatus* (Linnaeus) from the waters off Lizard Island on the northern Great Barrier Reef. We present a molecular phylogenetic analysis of the position of this genus inferred from 28S rDNA sequences. Surprisingly, we find the species most closely related to *Echeneidocoelium indicum* despite the infection of completely unrelated hosts and the presence of two characters (lateral fold in the forebody and multiple testes) that are found elsewhere in the Lepocreadiidae. We conclude that homoplasy within the Lepocreadiidae is extensive and that morphology-based prediction of relationships has little prospect of success.

**Keywords:** Digenea; Lepocreadiidae; *Deraiotrema*; *Platax*; Great Barrier Reef; phylogeny; homoplasy

## 1. Introduction

Recent molecular analyses have revolutionised the classification of the Lepocreadioidea. The process started with a molecular exploration of the superfamily [1] which identified five major clades and that the generally accepted concept of the Lepocreadiidae was polyphyletic. These issues were partly resolved by Bray & Cribb [2], by the recognition of Aephnidiogenidae Yamaguti, 1934 and Lepidapedidae Yamaguti, 1958 at the family level. At that stage, nine lepocreadioid genera were considered unplaceable (*genera incertae sedis*). Most recently, in Bray et al. [3], one of these genera, *Gibsonivermis* Bray, Cribb & Barker, 1997, was shown to justify recognition of a new monotypic family with the Lepocreadioidea, the Gibsonivermidae. Although we suspect that the overall structure of the Lepocreadioidea is maturing, there remains considerable uncertainty about the relative position of many of the genera. From this it follows that we have limited understanding of character evolution in any of the families. Here we redescribe an enigmatic genus and species, *Deraiotrema platacis* Machida, 1982, and produce an expanded phylogenetic analysis using 28S DNA, allowing consideration of the distribution of its distinctive defining characters.

## 2. Materials and Methods

### 2.1. Specimen Collection and Morphological Analysis

Fish were obtained by spear-fishing from the waters around Lizard Island, Queensland, Australia (14°40′S, 145°28′E). Trematodes were collected as described by Cribb & Bray [4]. Specimens were

pipetted into almost boiling saline for fixation and preserved in formalin or 70% ethanol. Whole-mounts were stained with Mayer's paracarmine or Mayer's haematoxylin, dehydrated in a graded ethanol series, cleared in beechwood creosote or methyl salicylate and mounted in Canada balsam. Measurements were made through a drawing tube on an Olympus BH-2 microscope, using a Digicad Plus digitising tablet and Carl Zeiss KS100 software adapted by Imaging Associates, and are quoted in micrometers, with the range and the mean in parentheses. The following abbreviations are used: NHMUK, the Natural History Museum, London, UK; QM, Queensland Museum Collection, Brisbane, Australia.

*2.2. Molecular Sequencing and Phylogenetic Analysis*

The protocols for molecular analysis were as described by Cutmore et al. [5] and Wee et al. [6]. The complete ITS2 rDNA region was amplified and sequenced using the primers 3S [7] and ITS2.2 [8], the partial D1-D3 28S rDNA region using LSU5 [9], 300F [10], ECD2 [11] and 1500R [12] and the partial *cox*1 mtDNA region using Dig_cox1Fa [6] and Dig_cox1R [6]. Assembly and editing of contiguous sequences utilised the Geneious® version 10.2.3 [13] and the start and end of the ITS2 rDNA region were determined by annotation through the ITS2 Database [14,15] using the 'Metazoa' model.

MUSCLE version 3.7 [16] run on the CIPRES portal [17] was used to align the partial 28S rDNA sequences generated during this study with sequences of other lepocreadiids derived from GenBank, with ClustalW sequence weighting and UPGMA clustering for iterations 1 and 2. The alignment obtained was revised by eye using MESQUITE [18]. The sequence ends were trimmed and ambiguously aligned regions were identified and masked manually (those with more than three bases and in greater than 5% of the sequences). Bayesian inference analysis of the 28S dataset was implemented using MrBayes version 3.2.6 [19] and maximum likelihood analysis using RAxML version 8.2.10 [20], both run on the CIPRES portal. The best nucleotide substitution model was estimated using jModelTest version 2.1.10 [21]. The Akaike Information Criterion (AIC) and Bayesian Information Criterion (BIC) predicted the GTR+I+Γ model as the best estimator; Bayesian inference and maximum likelihood analyses were performed using the closest approximation to this model. Nodal support in the maximum likelihood analysis was assessed by performing 100 bootstrap pseudoreplicates. Bayesian inference analysis was run over 10,000,000 generations (ngen = 10,000,000) with two runs each containing four simultaneous Markov Chain Monte Carlo (MCMC) chains (nchains = 4) and every 1000th tree saved. Bayesian inference analysis used the following parameters: "nst = 6", "rates = invgamma", "ngammacat = 4", and the priors parameters of the combined dataset were set to "ratepr = variable". Samples of substitution model parameters, and tree and branch lengths were summarized using the parameters "sump burnin = 3000" and "sumt burnin = 3000". Species of the family Aephnidiogenidae Yamaguti, 1934 and Gorgocephalidae Manter, 1966 were chosen as functional outgroup taxa, following Bray et al. [3].

**Genus *Deraiotrema* Machida, 1982**

***Deraiotrema platacis* Machida, 1982** (Figure 1)
**Type- host:** *Platax orbicularis* (Forsskål) Ephippidae, Orbicular batfish.
**Type-locality:** Waters around Palau.
**New Host:** *Platax pinnatus* (Linnaeus), Ephippidae, Dusky batfish.
　　**Site**: Intestine.
　　**Locality**: Coconut Beach, Lizard Island, Queensland, Australia (14°40′S, 145°28′E, 25/04/2015).
　　**Prevalence:** 1 of 7 (14%).
　　**Vouchers**: QM G238136–43, 8 wholemounts, including 2 hologenophores; NMHUK 2019.6.14.1-3, 3 wholemounts.
　　**Molecular data**: ITS2 rDNA, two identical replicates (one submitted to GenBank MN073840); cox1 mtDNA, two identical replicates (one submitted to GenBank MN073842); 28S rDNA, two identical replicates (one submitted to GenBank MN073841).

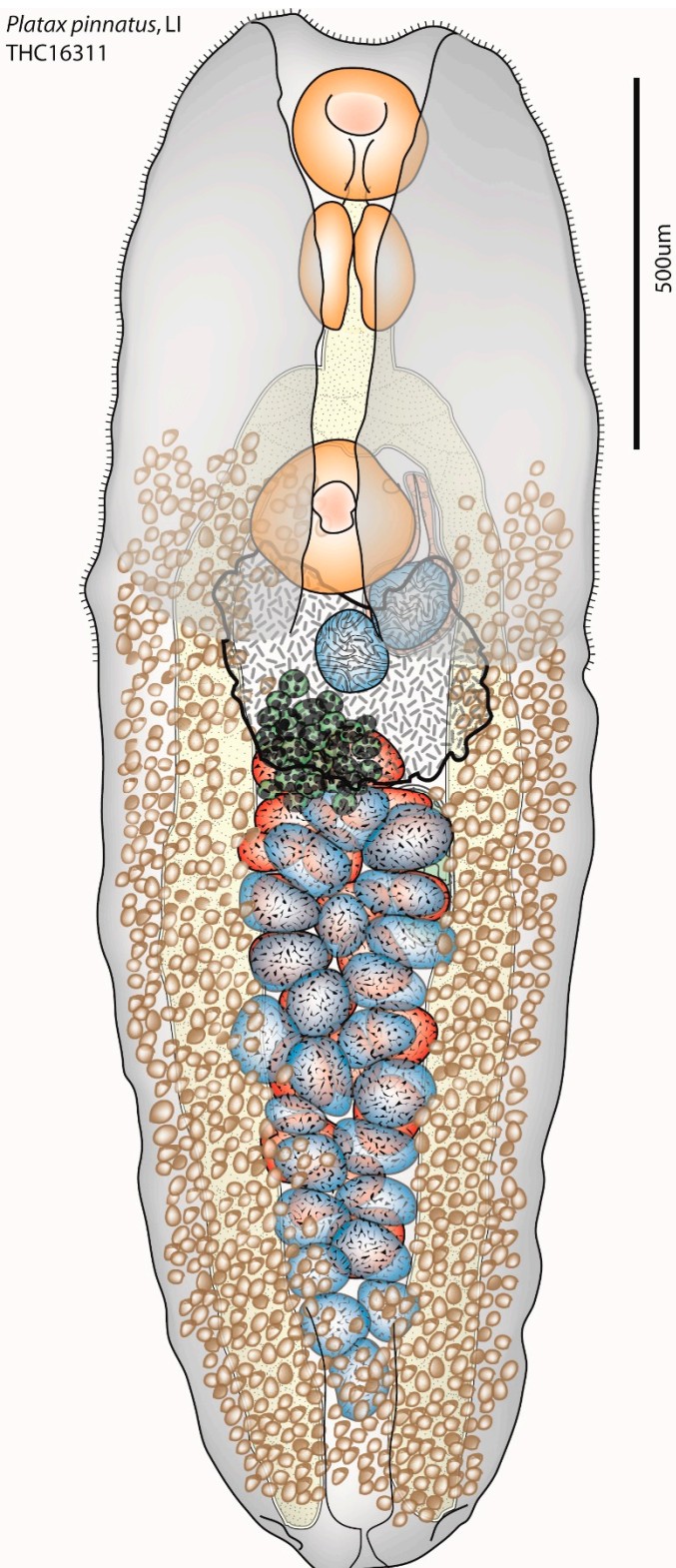

**Figure 1.** *Deraiotrema platacis* Machida, 1982. Ventral view, testes in ventral field blue, dorsal field red. Scale bar 500 μm.

*Description*: Measurements are of three ovigerous worms. Eight immature specimens were also available. Body elongate oval, 1838–2025 × 604–674 (1906 × 639), width 32.9–34.5 (33.5%) of body-length; with distinct lateral flaps folded over forebody and ventral sucker. Tegument finely spined to anterior

hindbody. Pre-oral lobe distinct, 25–37 (32). Oral sucker rounded, aperture ventral, 170–178 × 179–188 (174 × 184). Ventral sucker rounded, larger than oral sucker, 203–206 × 216–229 (205 × 222), 497–553 (529) from anterior extremity. Forebody 27.0–28.9 (27.8%) of body length. Oral-to-ventral sucker length ratio 1:1.14–1.21 (1.17); oral-to-ventral sucker width ratio 1:1.17–1.22 (1.20). Prepharynx short, within posterior concavity of oral sucker. Pharynx oval, large, of similar size to oral sucker; 178–184 × 156–163 (182 × 160). Pharynx-to-oral sucker width ratio 1.14–1.17 (1.15). Oesophagus distinct, 37–55 (45) long, 2.00–2.73 (2.36%) of body-length, 7.41–9.99 (8.50%) of forebody length. Intestinal bifurcation in posterior forebody, 64–92 (77) from ventral sucker, 3.51–4.97 (4.05%) of body-length, 13.0–17.2 (14.5%) of forebody length. Caeca broad, abut indented body wall posteriorly, possibly with ani.

Testes numerous, 38–44 (40.8), in two clearly distinct layers [ventral layer with 20–24 (21.8), dorsal layer with 17–21 (19.8)], small, oval, 105–120 × 72–92 (110 × 83); testicular fields in about middle two thirds of hindbody, almost completely intercaecal, fields 620–829 × 272–285 (698 × 279); post-testicular distance 193–268 (229), 9.55–14.5 (12.1%) of body-length. External seminal vesicle subglobular, in anterior hindbody in only mature specimen where its extent is not obscured. Cirrus-sac claviform, widest proximally, mostly sinistral to ventral sucker and reaching just into hindbody, 340–376 × 119–129 (358 × 124), length 16.8–20.3 (18.5%) of body-length. Internal seminal vesicle oval. Pars prostatica small. Ejaculatory duct strait. Genital atrium small. Genital pore close to antero-sinistral margin of ventral sucker.

Ovary multilobate, 156–192 × 174–202 (178 × 191), at level of anterior extent of testicular field, 89–98 (94), or 4.72–5.31 (4.94%) of body-length, from ventral sucker. Details of proximal female system obscured by testes. Uterus mostly in region just posterior to ventral sucker and overlapping ovary and testicular fields posteriorly. Eggs numerous, tanned, operculate, 64–73 × 36–39 (67 × 37). Vitellarium follicular, lateral fields overlapping caeca dorsally and ventrally, confluent only in post-testicular region and posterior-most part of testicular field, pre-vitelline distance 472–526 (498), 25.4–27.0 (26.1%) of body-length; anterior vitelline extent reaching 0–63 (31) into forebody, 0–3.49 (1.62%) of body-length, 0–12.1 (5.70%) of forebody length; posterior vitelline extent close to posterior extremity.

Excretory pore terminal. Excretory vesicle narrow posteriorly, widens abruptly and passes to testicular field where it can no longer be traced.

## 3. Molecular Results

Alignment of the new sequence data with that of related taxa (Table 1) yielded 1302 characters (including indels). Deleted ambiguously aligned regions amounted to 12 bases (<1% of the alignment), resulting in a final dataset of 1290 characters for phylogenetic analysis. Bayesian inference and maximum likelihood analyses resulted in phylograms with identical topologies (Figure 2), in which the species of the Lepocreadiidae formed a well-supported clade. Within the Lepocreadiidae, *D. platacis* formed a well-supported clade with *Echeneidocoelium indicum* Simha & Pershad, 1964, which together formed a well-supported clade with three species of *Hypocreadium* Ozaki, 1936. Notably, *D. platacis* was phylogenetically distinct from members of the *Pelopscreadium/Diploproctodaeum/Lobatocreadium/Diplocreadium/Bianium* clade, which are most similar to *D. platacis* morphologically.

**Table 1.** Collection data and GenBank accession numbers for lepocreadioid species analysed in this study.

| Species | Host Species | GenBank Accession # | Reference |
|---|---|---|---|
| **Lepocreadiidae Odhner, 1905** | | | |
| *Bianium arabicum* Sey, 1996 | *Lagocephalus lunaris* (Bloch & Schneider) | MH157076 | [3]) |
| *Bianium plicitum* (Linton, 1928) Stunkard, 1931 | *Torquigener pleurogramma* (Regan) | MH157066 | [3] |
| *Clavogalea trachinoti* (Fischthal & Thomas, 1968) Bray & Gibson, 1990 | *Trachinotus coppingeri* Günther | MH157067 | [3] |
| *Deraiotrema platacis* Machida, 1982 | *Platax pinnatus* (Linnaeus) | XXXXXX | Current study |
| *Diplocreadium tsontso* Bray, Cribb & Barker, 1996 | *Balistoides conspicillum* (Bloch & Schneider) | FJ788472 | [1] |
| *Diploproctodaeum momoaafata* Bray, Cribb & Barker, 1996 | *Ostracion cubicus* Linnaeus | FJ788474 | [1] |
| *Diploproctodaeum monstrosum* Bray, Cribb & Justine, 2010 | *Arothron stellatus* (Anonymous) | FJ788473 | [1] |
| *Diploproctodaeum* cf *monstrosum* | *Arothron hispidus* (Linnaeus) | MH157069 | [3] |
| *Echeneidocoelium indicum* Simha & Pershad, 1964 | *Echeneis naucrates* Linnaeus | FJ788475 | [1] |
| *Hypocreadium patellare* Yamaguti, 1938 | *Balistoides viridescens* (Bloch & Schneider) | FJ788478 | [1] |
| *Hypocreadium picasso* Bray, Cribb & Justine, 2009 | *Rhinecanthus aculeatus* (Linnaeus) | FJ788479 | [1] |
| *Hypocreadium toombo* Bray & Justine, 2006 | *Pseudobalistes fuscus* (Bloch & Schneider) | FJ788480 | [1] |
| *Lepidapedoides angustus* Bray, Cribb & Barker, 1996 | *Epinephelus cyanopodus* (Richardson) | FJ788482 | [1] |
| *Lepotrema adlardi* (Bray, Cribb & Barker, 1993) Bray & Cribb, 1996 | *Abudefduf bengalensis* (Bloch) | MH730015 | [22] |
| *Lepotrema melichthydis* Bray, Cutmore & Cribb, 2018 | *Melichthys vidua* (Richardson) | MH730021 | [22] |
| *Lepotrema moretonense* Bray, Cutmore & Cribb, 2018 | *Prionurus microlepidotus* Lacépède | MH730023 | [22] |
| *Lobatocreadium exiguum* (Manter, 1963) | *Pseudobalistes fuscus* | FJ788484 | [1] |
| *Mobahincia teirai* Bray, Cribb & Cutmore, 2018 | *Platax teira* (Forsskål) | MH157068 | [3] |
| *Multitestis magnacetabulum* Mamaev, 1970 | *Platax teira* | MH157071 | [3] |
| *Neohypocreadium dorsoporum* Machida & Uchida, 1987 | *Chaetodon flavirostris* Günther | FJ788487 | [1] |
| *Neomultitestis aspidogastriformis* Bray & Cribb, 2003 | *Platax teira* | MH157072 | [3] |
| *Neopreptetos arusettae* Machida, 1982 | *Pomacanthus sexstriatus* (Cuvier) | FJ788490 | [1] |
| *Opechona austrobacillaris* Bray & Cribb, 1998 | *Pomatomus saltatrix* Linnaeus | MH157073 | [3] |
| *Opechona kahawai* Bray & Cribb, 2003 | *Arripis trutta* (Forster) | FJ788491 | [1] |
| *Pelopscreadium spongiosum* (Bray & Cribb, 1998) Dronen, Blend, Khalifa, Mohamadain & Karer, 2016 | *Ostracion cubicus* | FJ788469 | [1] |
| *Preptetos caballeroi* Pritchard, 1960 | *Naso vlamingii* (Valenciennes) | AY222236 | [23] |
| *Preptetos trulla* (Linton, 1907) Bray & Cribb, 1996 | *Ocyurus chrysurus* (Bloch) | AY222237 | [23] |
| *Prodistomum keyam* Bray & Cribb, 1996 | *Monodactylus argenteus* (Linnaeus) | MH157074 | [3] |
| **Outgroup taxa** | | | |
| **Aephnidiogenidae Yamaguti, 1934** | | | |
| *Aephnidiogenes major* Yamaguti, 1934 | *Diagramma pictum labiosum* (Macleay) | FJ788468 | [1] |
| *Austroholorchis sprenti* (Gibson, 1987) Bray & Cribb, 1997 | *Sillago maculata* Quoy & Gaimard | MH157075 | [3] |
| *Holorchis castex* Bray & Justine, 2007 | *Diagramma pictum pictum* (Thunberg) | FJ788476 | [1] |
| **Gorgocephalidae Manter, 1966** | | | |
| *Gorgocephalus kyphosi* Manter, 1966 | *Kyphosus vaigiensis* (Quoy & Gaimard) | AY222234 | [23] |
| *Gorgocephalus yaaji* Bray & Cribb, 2005 | *Kyphosus cinerascens* (Forsskål) | KU951489 | [24] |

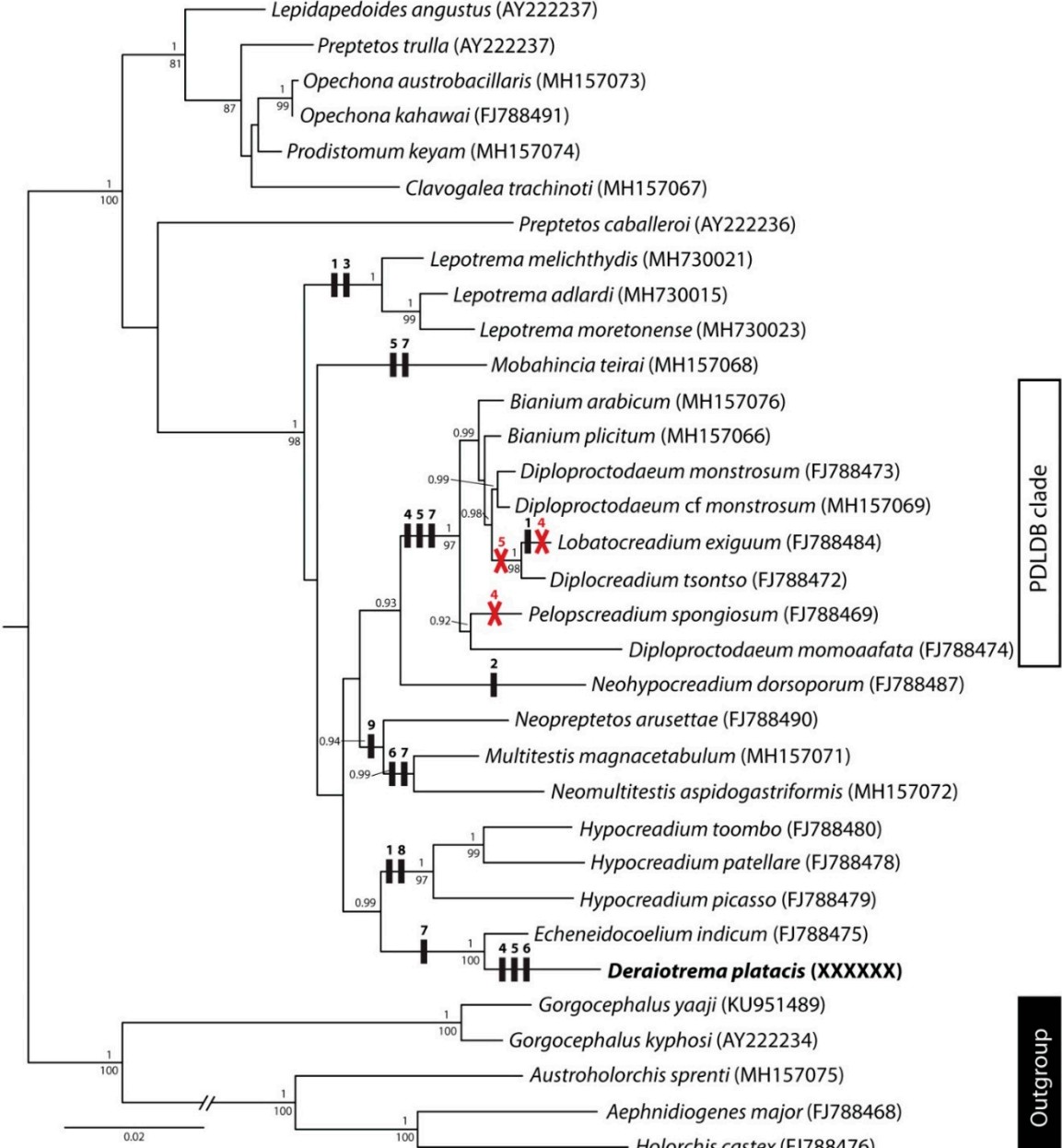

**Figure 2.** Relationships between members of the Lepocreadiidae based on maximum likelihood analysis of the partial 28S rDNA dataset. Homoplasious characters are numbered on the phylogram; black vertical bar indicates the acquisition of a character, a red cross the loss of character or reacquisition of the ancestral state. 1, Distinctly dorsal excretory pore; 2, Distinctly dorsal genital pore; 3, Muscular metraterm bulb; 4, Folds/flanges in forebody; 5, Caeca abutting body wall (? ani formed); 6, Multiple testes; 7, Multilobate/acinous ovary; 8, Flattened circular body; 9, Uterus reaches posterior to gonads. Bayesian inference posterior probabilities are shown above the nodes and maximum likelihood bootstrap support values below. Support values <0.80 and <80 are not shown. *Abbreviations*: PDLDB clade, *Pelopscreadium/Diploproctodaeum/Lobatocreadium/Diplocreadium/Bianium* clade.

## 4. Discussion

These worms appear to be indistinguishable from the only previously known specimens described by Machida [25] from the orbicular batfish *Platax orbicularis* from the waters off Palau, Micronesia. This is thus the second record of this species. Any slight differences between the original description and

our specimens probably relate to differences in fixation, with the specimens of Machida flattened in contrast to our new specimens.

*Deraiotrema* is distinguished by the combination of multiple testes, body flaps in the anterior part of the body, and caeca which abuts the body wall, giving the impression that ani may be present. Each of these characters is seen in other lepocreadiids, but the combination is unique.

Species of several lepocreadiid genera have anterior flaps or scoop-like developments of the anterior part of the worm, namely *Bianium*, *Diploproctodaeoides* Reimer, 1981, *Diploproctodaeum* and *Diplocreadium* Park, 1939 [26]. Strikingly, several species of these genera also infect ephippids, leading us, mistakenly, to expect that *D. platacis* would be related to them. Notably, none of these have multiple testes although the first three genera have caeca abutting the body wall, possibly forming ani. Other genera have a similar arrangement of the caeca, but no body-flaps, i.e., *Pelopscreadium* Dronen, Blend, Khalifa, Mohamadain & Karar, 2016 and *Mobahincia* Bray, Cribb & Cutmore, 2018 [3,27], neither of which have multiple testes. *Diploproctia* Mamaev, 1970 was illustrated with a scoop by Bray [26], but Machida [28] redescribed the type- and only species *D. drepanei* Mamaev, 1970 showing that Bray's interpretation of the illustration in Mamaev [29] was in error and the species lacks a scoop. Mamaev [29] stated that separate ani are present in *D. drepanei* but Machida [28] found no evidence for this and noted that the caeca terminate 'blindly near posterior end of body'. Several lepocreadiid genera have multiple testes, namely *Neomultitestis* Machida, 1982, *Transversocreadium* Hafeezullah, 1970, *Multitestis* Manter, 1931, *Rhagorchis* Manter, 1931 and *Multitestoides* Yamaguti, 1971 [26]. Perhaps the most similar genus to *Deraiotrema* is *Rhagorchis*, for which just two species are known, *R. odhneri* Manter, 1931 (syn. *R. varians* (Linton, 1940)) and *R. manteri* Ramadan, 1982 [30–32]. Yamaguti [33] re-drew the holotype of *R. odhneri* and showed that the ventral sucker is 'often retracted in anterior half of body'. The illustration by Yamaguti shows this is not the same feature as the lateral flaps we found on our specimens, although Yamaguti's illustration of this feature does resemble that in the drawing of *D. platacis* in Machida [25], where the lateral flaps are said to join posterior to the ventral sucker. We could not detect if the flaps join in our specimens and postulate, again, that the difference is due to flattening. The drawings of Manter [30] and Yamaguti [33] show the caeca in *R. odhneri* reaching fairly close to the posterior body wall but they do not appear to abut it and it appears that there are 11 testes in a single layer. *Rhagorchis odhneri* is known only from the monacanthids the orange filefish *Aluterus schoepfii* (Walbaum), the scrawled filefish *A. scriptus* (Osbeck) and the fringed filefish *Monacanthus ciliatus* (Mitchill) in the north-western Atlantic and the Caribbean Sea/Gulf of Mexico area [30,31,34–38]. The similarities between the species of *Deraiotrema* and *Rhagorchis* may be superficial, such that we would be tempted to think that *R. odhneri* is actually a schistorchiine apocreadiid were it not for the cirrus-sac in the re-drawing of the holotype by Yamaguti [33]; several species of that family with multiple testes are known from monacanthids. *Rhagorchis manteri*, from the scarid the longnose parrotfish *Hipposcarus harid* (Forsskål) in the Red Sea [32], is also likely to be a schistorchiine apocreadiid, possibly *Plesioschistorchis haridis* (Nagaty, 1957), which is also from scarids in the Red Sea [39,40]. Saoud et al. [41] reported *Rhagorchis* sp. in the bluechin parrotfish *Scarus ghobban* Forsskål, also from the Red Sea.

In phylogenetic analyses, *Deraiotrema platacis* formed a well-supported clade sister to *Echeneidocoelium indicum*, a parasite of remora *Remora remora* (Linnaeus) and, more usually, the sharksucker *Echeneis naucrates* Linnaeus. Morphologically and in terms of host-specificity, there is little to connect *D. platacis* and *E. indicum*. *Echeneidocoelium indicum* is a narrow, elongate worm, with two testes and caeca which impinge on the excretory vesicle, but apparently do not form a uroproct or ani [42,43]. One feature shared is the follicular ovary, a feature which is unusual in elongate lepocreadiids, but not in more squat forms. The intimate relationship with *E. indicum* is all the more surprising for the short branch lengths that separate them. The distance between them is less than that between any of the combinations of the three *Hypocreadium* species in the analysis. It is apparent, however, that features such as the anterior scoop and the appearance (or presence) of ani, and certainly the acquisition of multiple testes, are likely to be convergent features.

As can be seen from the new phylogram (Figure 2) the anterior scoop appears to be synapomorphic for the *Pelopscreadium/Diploproctodaeum/Lobatocreadium/Diplocreadium/Bianium* clade (PDLDB clade) but lost in *Pelopscreadium* and *Lobatocreadium;* we take this as further evidence that seemingly significant characters are highly labile. *Diploproctodaeum* as presently recognised by its complete anterior scoop (compared to lateral flaps only in *Bianium*) is not monophyletic and clearly the relationships of these genera need further study and wider sampling. It seems likely that the 'sponge-like pelops (shoulder pads)' in *Pelopscreadium* are homologues of the scoop or flaps [17,44]. *Lobatocreadium* Madhavi, 1972, represented in the tree by *L. exiguum* (Manter, 1963), has no obvious flap or scoop vestige [45–47]. Given the topology of the new phylogram, it seems unlikely that the lateral flaps in *Deraiotrema* are homologous with those of the PDLDB clade.

The appearance of ani in most members of the PDLDB clade is highly characteristic, with the caeca abutting the body-wall at the point of a distinct concavity in the wall. This feature is not, however, present in all members of the clade, with the caeca in species of *Lobatocreadium* and *Diplocreadium* clearly ending blindly well-separated from the body wall. The posterior terminations of the caeca in *Mobahincia* and *Deriaotrema* are very similar to those in the PDLDB clade but the condition must be homoplasious.

Multiple testes in two layers is a pattern which has been not described from any other lepocreadiid as far as we are aware. Apart from the multi-testiculate lepocreadiids mentioned above, multiple testes crop up scattered across the digenean tree. Many members of the superfamily Schistosomatoidea Stiles & Hassall, 1898 have multiple testes [48], as do members of the families Typhlocoelidae Harrah, 1922 [49], Syncoeliidae Looss, 1899 [50] and Orchipedidae Skrjabin, 1913 [51]. Sporadic genera in other families have multiple testes: Acanthocolpidae Lühe, 1906: *Pleorchis* Railliet, 1896 [52]; Opecoelidae Ozaki, 1925: *Decemtestis* Yamaguti, 1934, *Helicometrina* Linton, 1910 [53]; Cryptogonimidae Ward, 1917: *Acanthosiphodera* Madhavi, 1976, *Iheringtrema* Travassos, 1947, *Lobosorchis* Miller & Cribb, 2005, *Novemtestis* Yamaguti, 1942, *Polyorchitrema* Srivastava, 1939, *Siphodera* Linton, 1910 [54]; Monorchiidae Odhner, 1911: *Octotestis* Yamaguti, 1951 [55]; Gorgoderidae Looss, 1899: *Gorgotrema* Dayal, 1938, *Progorgodera* Brooks & Buckner, 1976, *Gorgodera* Looss, 1899, *Probolitrema* Looss, 1902, *Anaporrhutum* Brandes in Ofenheim, 1900, *Petalodistomum* Johnston, 1913 [56]. The ease with which digeneans may acquire multiple testis is shown by species within the cryptogonimid genus *Siphomutabilus* Miller & Cribb, 2013, where two of the four known species, *S. gurukun* (Machida, 1986) and *S. aegyptensis* (Hassanine & Gibson, 2005), have nine testes and the other two, *S. raritas* Miller & Cribb, 2013 and *S. bitesticulatus* Miller & Cribb, 2013, have just two [57]. Despite this difference in testes arrangement *S. gurukun* (nine testes) and *S. raritas* (two testes) were shown to form a strongly-supported clade in the cryptogonimid phylogeny [57].

In summary, *Deraiotrema* has a number of unusual lepocreadiid features; none are unique, although the combination of these features is. The evidence we have, therefore, is that the anterior scoop, caeca abutting the body wall, the follicular ovary and the multiple testes are all features readily homoplasiously derived from the more usual arrangements. We conclude that it should not be assumed that the many remaining unsequenced lepocreadiids genera will not surprise us in their phylogenetic positions. We note that eight of the nine lepocreadioid genera designated as *genera incertae sedis* by Bray & Cribb [2] remain unsequenced.

**Author Contributions:** Writing—original draft, R.A.B.; Writing—review & editing, S.C.C. and T.H.C.

**Funding:** This research received no external funding

**Acknowledgments:** We thank the staff at Lizard Island Research Station for their support in the field and Storm Martin who collected the host.

**Conflicts of Interest:** The authors declare no conflict of interest.

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
