# Peer review of "An Anomalous Phylogenetic Position for Deraiotrema platacis Machida, 1982 (Lepocreadiidae) from Platax pinnatus on the Great Barrier Reef"

_diversity, doi:10.3390/d11070104_

Round 1
Reviewer 1 Report
This paper provides a valuable redescription of Deraiotrema platacis as well as an analysis of the phylogenetic relations within Lepocreadiidae. I find this paper extremely welcome since it provides valuable insights into the evolution of the group, including a well supported discussion on the evolution of morphological characters within the group. The main conclusion is that homoplasy is recurrent within the group, casting doubts on the use of morphological characters to infer phylogenetic relations. This conclusion is applicable to many other parasite groups, especially those microscopic with very simple morphological features. Also for this reason I think this paper will be highly cited. I only have one minor comment. The authors amplified other genes besides 28S rRNA, the ITS and a portion of COI. However these results are not described or discussed, nor it seems that sequences will be deposited on Genbank (based on Table 1). I think this should be clarified in the text and the results presented, even if the phylogentic tree would be not as complete.
Reviewer 2 Report
The paper "An
anomalous phylogenetic position for Deraiotrema platacis Machida,
1982 (Lepocreadiidae) from Platax pinnatus on the Great
Barrier Reef" is an excellent study highlighting the importance
of phylogenetic analysis of trematode parasites. It combines a
thorough morphological description of the trematode Deraiotrema
platacis, which is recorded only
for the second time ever, with its phylogenetic analysis. If
identification had been based on morphology alone, this trematode,which infects batfishes, would be placed with a group of lepocreadiid
trematodes including Bianium and other trematode species that present folds or body flaps over the
forebody and that occur in similar host fishes. However, phylogenetic
analysis based on 28S rDNA clearly places the worms in a different
clade, more closely related to Echeneidocoelium indicum,
a parasite of remoras that is morphologically quite different (see
for example Madhavi 1970). As
the authors point out, this seems to indicate that several trematode
features are easily homoplasiously re-evolved. The need for
sequencing other lepocreadioid genera is highlighted at the end of
the paper.
In my opinion the paper can be published practically as is; there are no important errors to note. A few minor comments:
- Although usually absent in systematic papers, any information about infection levels, number of fish observed, etc would be useful to parasitologists with a moreepidemiological focus.
- line 233 "Iheringotrema Travassos, 1947" is given as "Iheringtrema Travassos, 1948" in the World Register of Marine Species, as well as in other studies, e.g. Fernandes & Kohn (2001). Which is correct?
Reference:
Fernandes, B. M. M., & Kohn, A. (2001). On some trematodes parasites of fishes from Paraná River. Brazilian Journal of Biology,61(3), 461-466.
-The authors could potentially extend the conclusions regarding lability of traits and frequency of homoplasy to a wider group of organisms than lepocreadiids, such as trematodesor even platyhelminthes in general. For instance, similar discrepancies between morphology and phylogenetics have been found in Capsalid monogeneans (Perkins et al. 2009).
Reference:
Perkins, E. M., Donnellan, S. C., Bertozzi, T., Chisholm, L. A., & Whittington, I. D. (2009). Looks can deceive: molecular phylogeny of a family of flatworm ectoparasites (Monogenea: Capsalidae) does not reflect current morphological classification. Molecular Phylogenetics and Evolution,52(3), 705-714.
- The repeated evolution of multiple testes in different trematode groups is not particularly surprising, as this trait is likely to confer some reproductive advantage in many contexts. But what could be the evolutionary significance, if any, of the lateral flaps in the forebody?
